# The STAIR OF KNOWLEDGE—a codesigned intervention to prevent pressure ulcers, malnutrition, poor oral health and falls among older persons in nursing homes in Sweden: development of a complex intervention

Merita Neziraj [1], Malin Axelsson,[1] Christine Kumlien,[1,2] Peter Hellman,[1] Magdalena Andersson[3]

For numbered affiliations see end of article.

**Correspondence to**
Merita Neziraj;
merita.neziraj@mau.se

## ABSTRACT

**Objectives** To describe the development of a codesigned complex intervention intended to prevent the risks of pressure ulcers, malnutrition, poor oral health and falls among older persons in nursing homes.

**Design** A complex intervention development study. The development of the intervention was conducted in three phases. We established contact with stakeholders in the municipality, updated us of current status of the literature in this area and conducted studies in the local context (1). We codesigned the intervention in workshops together with end users (2). We codesigned the final outline of the intervention in an iterative process with stakeholders (3).

**Setting**: Nursing homes in the municipality in southern Sweden.

**Participants** End users (n=16) in nursing homes (n=4) codesigned the intervention together with the research group in workshops (n=4) in March–April 2022. Additionally, stakeholders (n=17) who were considered to play an important role in developing the intervention participated throughout this process. Data were analysed using reflexive thematic analysis.

**Results** Four workshops were conducted with end users (n=16) and 13 meetings with stakeholders (n=12) were held during the development process. The intervention aims to bridge the evidence-practice gap regarding the preventive care process of the risks of pressure ulcers, malnutrition, poor oral health and falls among older persons in nursing homes. The intervention is aimed at end users, lasts for 3 weeks and is divided into two parts. First, end users obtain knowledge on their own by following written instructions. Second, they meet, interact and discuss the knowledge acquired during part 1.

**Conclusion** The intervention is robustly developed and thoroughly described. The study highlights the extensive process that is necessary for developing tailored complex interventions. The description of the entire development process may enhance the replicability of this intervention. The intervention needs to be tested and evaluated in an upcoming feasibility study.

**Trial registration number** NCT05308862.

### STRENGTHS AND LIMITATIONS OF THIS STUDY

⇒ Inspired by the Medical Research Council guidelines for complex intervention, a robust development process was undertaken based on the literature and research conducted in the local context prior to developing the complex intervention.

⇒ A complex intervention was codesigned both with and for nurse aides, registered nurses and managers in workshops. Additionally, key persons working in the municipality were engaged in the development of this tailored intervention.

⇒ To bridge the evidence–practice gap regarding the risks of pressure ulcers, malnutrition, poor oral health and falls among older persons in nursing homes, knowledge translation strategies were applied during the development process in accordance with the action part of the knowledge-to-action framework.

⇒ A thorough description of the entire development process may enhance the replicability of the current intervention.

⇒ One limitation of the development process was that this design is time-consuming and resource-consuming. On the other hand, this was necessary to develop a tailored complex intervention that might enhance the likelihood of successful implementation. The transferability of the tailored intervention to other nursing homes might also be a limitation.

## INTRODUCTION

There remains an evidence–practice gap in preventing the risks of pressure ulcers, malnutrition, poor oral health and falls among older persons in nursing homes.[1 2] These health risks cause a major burden for older persons[3] and they are costly for the healthcare system.[4] Since older persons are more vulnerable to these health risks[5] and considering the increasing ageing population

globally, particularly with regard to older persons aged 80 years or older,[6] evidence-based preventive work is crucial to manage this demographic challenge and, importantly, these health risks among older persons.

In Sweden, there is a national quality register, Senior Alert, providing an individualised, standardised, structured and systematic preventive care work process for older persons 65 years or older who are at risk of pressure ulcers, malnutrition, poor oral health and falls.[7] Senior Alert provides evidence-based knowledge aimed at preventing these health risks to enable a healthy ageing among older persons;[8] in addition, it can increase cost efficiency.[9] However, a lack of knowledge among those working with older persons has been identified as one major challenge regarding to preventive work.[2 10] As a result, these health risks continue to be prevalent.[7] For instance, approximately every third older person living in a nursing home faces at least one of these health risks, and every 10th older person faces all four of these health risks.[1] Additionally, not all older persons who are at risk have planned care interventions[11 12] and there is a mismatch between identified risks and planned and performed care interventions,[13 14] thus indicating an evidence–practice gap and consequently, highlighting the urgent need of translating knowledge into practice.

Nevertheless, this is not unique to Sweden or this context; in contrast, health systems worldwide face the shared challenge of translating knowledge into practice.[15] Knowledge translation has been defined as 'a dynamic and iterative process that includes synthesis, dissemination, exchange and ethically sound application of knowledge to improve health care of people in the country, provide more effective health service and products and strengthen the health care system', p. 165.[16] Ineffective knowledge translation can result in an evidence–practice gap[17] and, worryingly, lead to situations in which patients are denied interventions that have been proven to be beneficial,[18] which in turn can result in a reduction in their quality of life.[19]

To bridge this evidence–practice gap, conceptual frameworks are recommended.[20] The knowledge-to-action (KTA) framework is intended to help the parties involved in the process of knowledge translation.[18] The KTA framework is also appropriate when addressing an evidence–practice gap[15] and conducting pragmatic research.[18]

As a part of translating knowledge into practice and promoting knowledge use by end users,[21] the engagement of both researchers and stakeholders in research is crucial.[22] Engaging stakeholders at an early stage in the development of solutions that can be applied to real-world settings is essential according to the Medical Research Council's (MRC) framework for complex interventions.[23] Complex interventions have multiple components, target multiple groups or levels of an organisation and attempt to affect multiple outcomes.[23] Additionally, for complex interventions to be most useful to end users, the local context must be taken into account.[24] Since it

is well underpinned that organisational factors hinder preventive work in nursing homes,[2 25] considering and understanding the local context and integrating it into the process of intervention development is crucial.[26]

Consequently, change in the practices of nursing homes is considered to be complex,[27] but if complex interventions are tailored to the local context,[28] including the targets of the intervention[23 24] and is directly relevant to them,[29] such interventions could be successful.

## Aim
The aim of this study was to describe the development of a codesigned complex intervention intended to prevent pressure ulcers, malnutrition, poor oral health and falls among older persons in nursing homes.

## METHODS
### Definitions
*Nursing homes* were defined based on the definition provided by Neziraj *et al* (2021):[1] residential care homes where older persons live and receive municipal healthcare.

Healthcare personnel and managers were defined based on the definition provided by Neziraj *et al* (2021) as follows:[2]

*Nurse aide*: a person with a secondary degree in nursing involves 3 years of study in high school *or* a person without any formal education in nursing.

*Registered nurse*: a person with a bachelor's degree in nursing, which involves 3 years of study at university.

*Manager*: a person who is in charge of nurse aides or registered nurses.

*End users*: nurse aides, registered nurses and managers working in nursing homes.

*Stakeholders*: key persons working in the municipality who are considered to play an important role in the development and implementation of the intervention.

### Study context and setting
In nursing homes, nurse aides are the main providers of care and services and are on duty around the clock. Nurse aides work under the regulations of the Social and Services Act (SFS),[30] but are also delegated tasks according to the Health and Medical Services Act (HSL),[31] usually by registered nurses. Registered nurses guide care in nursing homes and work under the regulations of HSL.[31] In the current setting, a large town located in southern Sweden with 39 nursing homes, one registered nurse (or occasionally more depending on the size of the nursing home) is located in the nursing home during office hours but is also available at any other time. Managers who are in charge of the care and services provided by the nurse aides are located at their respective nursing homes during office hours.

For transparency, the research group (n=5) positions are reported; four of the researchers hold positions as either doctoral students (MN), associated professors

(MAx), professors (CK) or senior lectures (PH) at the affiliated university. The last author (MA) is a PhD and holds the position as a research and development coordinator in the municipality where the study was conducted. All the authors are registered nurses, and two of them (MN and MA) specialise in elderly care and have worked in this context previously,

In addition, a reference group was created, which consisted of experts (n=7) drawn from the local context; nurse aide (n=1), managers in charge of nursing homes (n=2), head of managers in charge of registered nurses (n=1), development managers (n=2) and head of the nursing homes in the municipality (n=1).

## Study design

This study describes the development of a codesigned complex intervention and is a part of the PROSENIOR programme (https://mau.se/en/research/projects/prosenior/). This part of the PROSENIOR programme aims to develop, test and evaluate a codesigned complex intervention to prevent pressure ulcers, malnutrition, poor oral health and falls among older persons living in nursing homes in a two-arm pragmatic cluster randomised trial. The randomisation was conducted with the double aim to first develop the intervention and then to evaluate the feasibility in the nursing homes allocated to the intervention group. This study only reports on the intervention development part of this trial. The feasibility evaluation regarding, for example, recruitment and retention of nursing homes and randomisation procedure will be reported separately elsewhere. In this study, the randomisation aimed to invite end users allocated to the intervention arm to develop a codesigned complex intervention. The control arm was therefore not included in this study. The nursing home is the cluster and the unit of allocation. The nursing homes were randomised using a computerised programme (Excel) by MN to either intervention or control arm. MN informed the managers in the included nursing homes about allocation output. Due to the nature of the design, the cluster randomisation of nursing homes was unblinded to the nursing homes and the researchers (figure 1).

The development of the codesigned complex intervention (hereafter called the intervention) was conducted in three phases. The phases are described below. The development of the current intervention was conducted in a pragmatic paradigm as it is intended to work in a real-world setting;[29] this process was inspired by the MRC guidelines for complex interventions,[24] applied the KTA framework[18] and engaged end users and stakeholders in the process of codesign.[32]

We follow the guidance for reporting intervention development studies (GUIDED)[33] when describing the development of the intervention and the template for intervention description and replication (TIDieR) checklist and guide[34] when describing the intervention. We use 'development' to refer to the whole process of

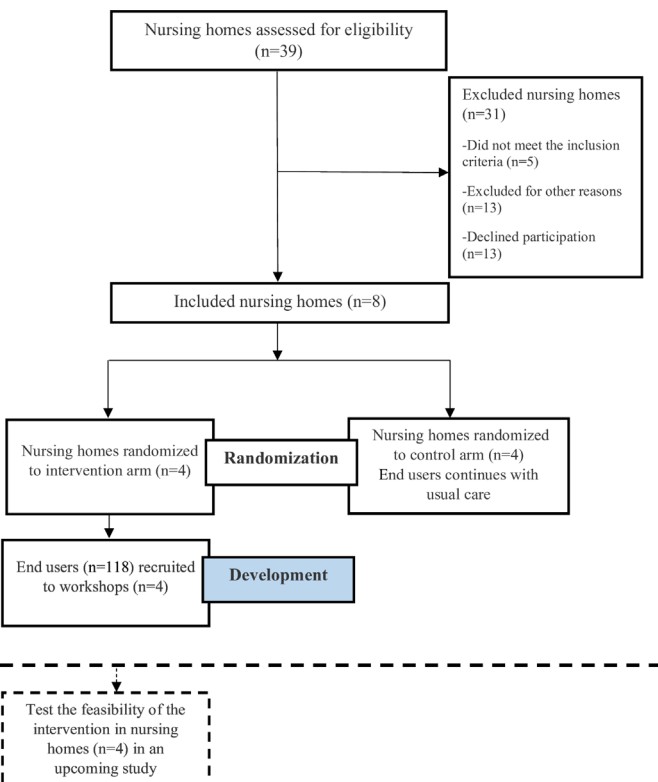

**Figure 1** Flowchart of the two-arm pragmatic cluster randomised controlled trial. The dashed lines illustrate an upcoming study. This study focused on the development of the STAIR OF KNOWLEDGE intervention.

intervention development and 'design' to indicate the intervention content, format and delivery.

## Patient and public involvement

Patients or informal caregivers were not involved in the research process. End users codesigned the intervention with the research group in workshops. Stakeholders were also involved in this research; they supported the research group throughout the entire development of the intervention by contributing their valuable knowledge. All engagement is described in detail in the section 'Development of the intervention' as follows.

## Development of the intervention

We developed the intervention in three phases and applied the KTA framework in all phases (figure 2).

### Theory

The KTA framework takes implementation strategies into account already in the development phase,[18] which promotes and sustains practice change.[15] We applied the KTA framework because it offers a structured and systematic approach to translate knowledge into practice.[18] It comprises two parts: knowledge creation and the action cycle. Since evidence-based knowledge is already available to end users in the quality register Senior Alert, the action cycle was applied during the development of the current intervention. The action cycle consists of the following steps: *(1) identify the problem, identify and review selected*

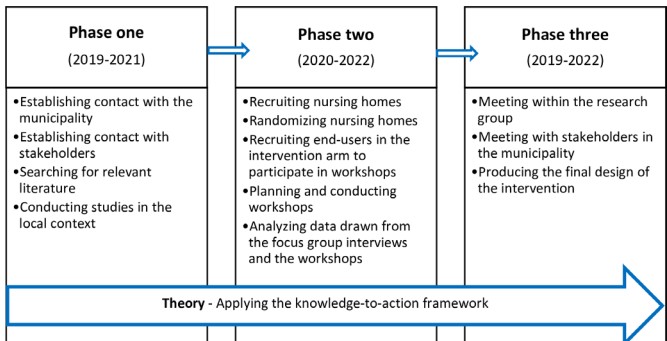

**Figure 2** Phases 1–3 illustrate the process of developing the STAIR OF KNOWLEDGE intervention, which took place between 2019 and 2022. Although the knowledge-to-action (KTA) framework is viewed as a cycle by Graham and colleagues (2006), in this figure, the arrow illustrates the fact that the KTA framework was applied throughout phases 1–3 of the development process. The KTA framework was applied in an iterative and dynamic way in each phase and is described in detail in the text.

knowledge, (2) adapt knowledge to the local context, (3) assess barriers to knowledge use, (4) select and tailor implementation strategies, (5) monitor knowledge use, (6) evaluate the outcomes and (7) sustain knowledge use.[18 35] Steps (1)–(4) the action cycle were applied throughout the development process of developing the intervention in an iterative, dynamic and permeable way.

### Phase 1
During this phase, we established contact with stakeholders in the municipality, updated us of the current status of the literature in this area and conducted studies in the local context.

*Establishing contact with stakeholders in the municipality*
Initially, we established contact and met with the head of the nursing homes in the municipality. The reference group was created in this phase (described in the paragraph 'Study context and setting' above).

*Searching for literature and conducting studies in the local context*
As a part of step 1 in the KTA framework, *identify the problem, identify and review selected knowledge,* first, we updated us of the current status of the literature regarding prevention of pressure ulcers, malnutrition, poor oral health and falls and intervention studies in this area. Subsequently, we conducted a cross-sectional study to determine the prevalence of the risks of pressure ulcers, malnutrition, poor oral health and falls in nursing homes in southern Sweden.[1]

As a part of steps 2–3 in the KTA framework, *adapt knowledge to the local context and assess barriers to knowledge use,* we conducted focus group interviews (n=5) with end users (n=21) who worked in nursing homes to prevent pressure ulcers, malnutrition, poor oral health and falls.[2] The focus group interviews lasted between 63 and 106 min (mean 83 min). A detailed description of this study and its participants is provided in the works of Neziraj *et al.*[2]

Additionally, we asked the end users included in our previous study[2] how an optimal intervention could be designed to prevent the risks of pressure ulcers, malnutrition, poor oral health and falls among older persons in nursing homes. These particular data were targeted for this study. Hence, these data were not reported in our previous study, but are included, analysed and reported in our current study.

### Phase 2
During this phase, we recruited and randomised nursing homes. Subsequently, we invited end users in the intervention arm to participate in workshops, and planned and conducted the workshops. We also analysed the specific data regarding intervention design drawn from the focus group interviews (see the previous paragraph on phase 1 for clarification) and the workshops.

*Recruiting and randomising nursing homes*
In this part of the two-arm pragmatic cluster randomised trial, randomisation aimed to recruit end users in nursing homes allocated to the intervention arm to codesign an intervention together with the research group in workshops.

Inclusion criteria for the study were nursing homes working with and registered in the quality register Senior Alert. We recruited eligible nursing homes (n=21) to participate in the study via digital meetings. In total, eight nursing homes agreed to participate and were cluster randomised using a computerised programme to either the intervention (n=4) or control arm (n=4). Subsequently, we invited end users (n=118) working in nursing homes in the intervention arm to participate in workshops intended to develop a tailored intervention together with the research group; the invitations were extended both via a digital information video and in written form. The remaining end users (n=184) working in the nursing homes who were allocated to the control arm continued with their usual care routine.

*Conducting workshops*
As a part of steps 2–4 in the KTA framework, *adapt knowledge to the local context, assess barriers to knowledge use* and *select and tailor implementation strategies,* we conducted workshops with end users. In total, four workshops were conducted, which featured two nurse aides, one registered nurse and one manager in each workshop; the workshops were conducted over the course of 4 weeks (March–April 2022). The workshops were kept small to offer the end users the possibility of exhibiting activity and creativity.[36] The first author (MN) led the workshops together with one of the coauthors (all coauthors participated in one workshop each). The workshops were intended to serve as a place in which participants could learn together and discuss the design of the intervention in four different stations (table 1). The end users engaged in active discussion and wrote creative ideas and suggestions on the walls and the board in a lecture hall designed for the

**Table 1** Workshop content (n=4)

| Workshop | Station | Content | Examples of questions to discuss |
|---|---|---|---|
| Workshop 1–4 | Station 1 | Case regarding an older person at risk of pressure ulcers, malnutrition, poor oral health and falls living in a nursing home | ▶ What would you have done in this case regarding these four risks?<br>▶ Are there any good examples? What can you learn from good examples?<br>▶ What additional knowledge do you need regarding these four risks in order to produce a risk assessment and provide adequate care interventions? |
| | Station 2 | Senior Alert's care process | ▶ Place green/pink post-it notes on the care process regarding what works/what can be improved in your own work and workplace. |
| | Station 3 | End users needs' and the support they need regarding preventive work | ▶ What do you need in your preventive work?<br>▶ Why is this important, and what is most important (ranks 1–3)?<br>▶ Who needs help in the context of preventive work?<br>▶ Who should be involved and in what way?<br>▶ What is necessary for it to be feasible?<br>▶ How can you work better/smarter?<br>▶ How can you work in a more sustainable way? |
| | Station 4 | Core components of the intervention | ▶ What should be included in the intervention?<br>▶ Who should it target?<br>▶ How should it be designed?<br>▶ How much/often/for how long should the intervention take place?<br>▶ How should it be followed up?<br>▶ Where should it be implemented?<br>▶ How should it be implemented? |

purpose of encouraging creative pedagogy. In the first station, the end users were asked to discuss the risks of pressure ulcers, malnutrition, poor oral health and falls and the care interventions that should be applied. In the second station, they were asked to discuss and identify barriers and facilitators they had encountered in their own work regarding the preventive care process stipulated by Senior Alert (identify a risk, assess causes and plan, undertake and evaluate care intervention). Barriers were written down on pink post-it notes, while facilitators were written down on green post-it notes. These post-it notes were subsequently placed at the appropriate location on the board with regard to the predawn preventive care process. The focus of the discussions at third station was on the end users' needs and the support they needed throughout the preventive care process. In the fourth station, they were asked to discuss the core components of the intervention, how to provide follow-ups and implementation strategies. After completing each workshop, MN photographed and briefly summarised the written data from each station. This summary was used if the end users in the subsequent workshop reached an impasse and/or discussed and wrote similar suggestions and ideas to those proposed by the end users in the previous workshop. Each workshop lasted for 3 hours, and the discussions were audio recorded to support the written data collection during the analysis.

### Analysing the data from the focus group interviews and the workshops

The analysis was guided by the six phases of reflexive thematic analysis described by Braun and Clarke:[37 38] *(1) familiarising with the data, (2) coding, (3) generating initial themes, (4) reviewing the identified themes, (5) defining and naming the themes* and *(6) producing the report.* Thematic analysis was chosen because it facilitates a flexible analysis process but simultaneously provides researchers with the core skills they need to conduct the analysis.

To familiarise ourselves with the data, MN and MA read the transcripts from the focus group interviews, including the data specifically collected for this study, and the written data collected from the workshops. In addition, MN listened to all the audio-recorded discussions from the workshops meticulously. During the process of

reading the data, MN and MA reflected on and generated initial codes. Subsequently, MN and MA met and discussed these initial codes (1). Thereafter, MN and MA separately engaged in a process of identifying and coding entities of interest in relation to the design of the intervention, giving equal attention to all the data (2). The initial codes were then sorted into their core components in relation to the design of the intervention (3). Next, the core components were reviewed by MN to determine whether any relevant data regarding the design of the intervention had been missed (4). Subsequently, MN designed an outline of the intervention. This outline contained the intervention's proposed design, including its content, format, plan for delivery and duration. In the following step of the analysis, the entire research group met and discussed the design of the outline of the intervention. During this step, MN continuously revised the outline of the intervention following discussions within the research group (5). Then, the outline of the intervention was redesigned by MN. The redesigned outline of the intervention was then presented to the research group before it was presented to the stakeholders. The process of producing the final design of the intervention is described in phase 3 as follows (6).

### Phase 3

As part of steps 2–4 in the KTA framework, *adapt knowledge to the local context, assess barriers to knowledge use* and *select and tailor implementation strategies,* MN and MA met regularly with stakeholders in structured meetings to present and discuss the outline of the intervention. MN documented all the meetings. MA works within the municipality and thus facilitated contact with stakeholders who were considered to play an important role in this part of designing the intervention. Since this part of the process was dynamic and iterative and because all relevant uncertainties had not been addressed in the redesigned outline of the intervention, it was helpful to meet stakeholders for the purpose of identifying and addressing the remaining uncertainties regarding the content, format, delivery and duration of the intervention. This part of the process was time-consuming and required a back-and-forth process involving meetings and discussions between MN and MA, within the entire research group and with the stakeholders. Next, the redesigned outline of the intervention was adjusted by MN in accordance with the results of these meetings and discussions (figure 3). Finally, MN investigated whether any data from the focus group interviews and the workshops had been missed, since these data were intended to serve as the foundation for designing the final outline of the intervention. The final design of the intervention, the STAIR OF KNOWLEDGE (figure 4), is described as follows.

### RESULTS

Findings from our previous studies[1] [2] in phase 1 showed that the prevalence of the risk for pressure ulcer, malnutrition, poor oral health and falls is still high in the local context. Furthermore, findings from phase 1 suggested that individuals working with older persons in nursing homes need increased knowledge concerning how to prevent these health risks. Since existing evidence and knowledge concerning how to prevent these health risks is already contained in Senior Alert, the challenge seems to lie in the evidence–practice gap. Consequently, in phases 2 and 3, a tailored intervention was codesigned with end users, stakeholders and the research group to reduce the evidence–practice gap. The final design of the intervention is presented below.

A majority of the end users (n=16) in workshops (n=4) were women (n=13), between the ages of 28–63 years (mean 53) and had worked for 3–41 years (mean 18). The meetings (n=13) with stakeholders (n=12) lasted between 60 and 180 min (mean 134 min).

### The final design of the intervention

The final design of the intervention was described in line with the template for intervention description and replication checklist[34] (online supplemental file).

The STAIR OF KNOWLEDGE consists of *the foundation and stairs 1–6,* lasts for 3 weeks and is divided into

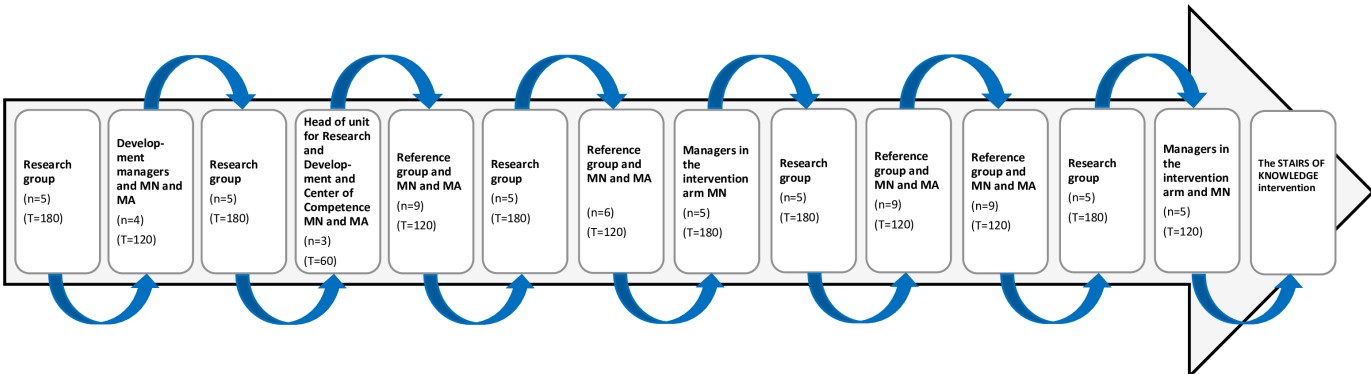

**Figure 3** The iterative and dynamic process of designing the final outline of the STAIR OF KNOWLEDGE intervention between April and September 2022, including meetings and discussions with stakeholders. In all the meetings, the first author participated. In addition, in some meetings also the last author participated. The blue arrows illustrate that adjustments were made following each meeting. Note: T=how long the meeting lasted for, reported in min. MN, the first author. MA, the last author.

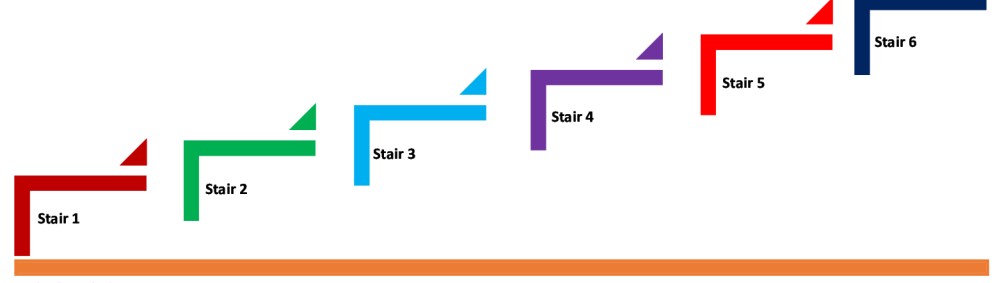

# The STAIR OF KNOWLEDGE

The STAIR OF KNOWLEDGE is addressed to all nurse aides, registered nurses and managers who work with older persons in nursing homes. The STAIR OF KNOWLEDGE aims to increase knowledge regarding the preventive care necessary to prevent the risks of falls, pressure ulcers, malnutrition and poor oral health.

Stair 1
Stair 2
Stair 3
Stair 4
Stair 5
Stair 6

The foundation

---

**The foundation** is mandatory for nurse aides, registered nurses and managers. **Stairs 1–4** and **stair 6** are mandatory for all nurse aides, registered nurses and managers. **Stair 5** is mandatory for individuals who register in Senior Alert. Follow the instructions below.

---

**For your convenience,** click the boxes as you progress through the STAIR OF KNOWLEDGE.

Foundation ☐    Stair 4a–d ☐
Stair 1a–d ☐    Stair 5 ☐
Stair 2a–d ☐    Stair 6 ☐
Stair 3 ☐

**The foundation.** Local working description of the entire preventive care working process.

- Link to the local working routine.

**Stair 1a–d.** General information regarding falls, pressure ulcers, malnutrition and poor oral health.

- Links to texts and videos regarding falls, pressure ulcers, malnutrition and poor oral health.

**Stair 2a–d.** Risk assessment of falls, pressure ulcers, malnutrition and poor oral health.

- Links to texts and videos regarding the risk assessment of falls, pressure ulcers, malnutrition and poor oral health.

**Stair 3.** Causes of falls, pressure ulcers, malnutrition and poor oral health.

- Link to text regarding the causes of falls, pressure ulcers, malnutrition and poor oral health.

**Stair 4a–d.** Preventive care interventions for falls, pressure ulcers, malnutrition and poor oral health.

- Links to texts regarding preventive care interventions for falls, pressure ulcers, malnutrition and poor oral health.

**Stair 5.** Registering in Senior Alert.

- Links to texts and videos regarding how to register in Senior Alert.

**Stair 6.** Inspiration week.

- Inspiration week focuses on preventive care intended to prevent the risks of falls, pressure ulcers, malnutrition and poor oral health in an inspiring and motivating way. The inspiration week will be organized by and for employees and managers. The inspiration week is preferably organized twice per year.

**Figure 4**  Final design of the STAIR OF KNOWLEDGE intervention.

two parts. Part 1, including *the foundation and stairs 1–5*, takes place throughout the entire intervention period (weeks 1–3) and is delivered digitally to end users in the nursing homes via their workplace email addresses. Part 2 includes *stair 6* and takes place during the last week of the intervention period (week 3) in the nursing homes in question (figure 4).

### The content of the intervention
#### Part one of the intervention: the foundation and stairs 1–5
End users emphasised uncertainties of different professionals' responsibilities regarding the preventive work. For instance, they highlighted that it is highly relevant for respective professional to know 'who does what' regarding the preventive working. Hence, *the foundation* is intended to facilitate for different professionals regarding responsibilities for respective profession and working routine in the local context. *The foundation* provide end users with knowledge and awareness of how to work preventively in the context of an existing local working routine and is intended to represent 'the ground to stand on'.

Furthermore, end users expressed a need of increased knowledge regarding the health risks and the entire preventive working process. They stressed the importance of basic knowledge when working with older persons in nursing homes. According to end users, not all of them has basic knowledge in how to prevent these health risks among older persons. This was particularly common among temporary workers. To meet their need, *stairs 1–4* provide the end users with general knowledge about risks of pressure ulcers, malnutrition, poor oral health and falls according to the care process suggested by the quality register Senior Alert (*stair 1),* risk assessment instruments (*stair 2),* the underlying causes of these risks (*stair 3*) and preventive care interventions (*stair 4). Stairs 1–5* provide end users with website links that allow them to both read texts and watch videos. *Stairs 1–4* are mandatory for all professionals. *Stair 5* provides end users with knowledge of how to register in the quality register Senior Alert and is mandatory only for users who have access to and the responsibility to register in the quality register Senior Alert.

#### Part two of the intervention: stair 6
Although it was necessary for end users to increase their knowledge on their own regarding the preventive work, they particularly highlighted the importance of physical meetings. This was also stressed as important by stakeholders and was considered as a complement to the first part of the intervention. Therefore, in part 2, *stair 6,* a facilitator (MN) meets with end users to interact with them and discuss the knowledge acquired during part 1. The meetings will be structured including discussions based on different cases related to pressure ulcers, malnutrition, poor oral health and falls. End users will also perform risk assessments, identify the underlying causes and plan accurate care interventions based on these cases. Additionally, end users will identify environmental risk factors related to the risks of pressure ulcers, malnutrition, poor oral health and falls in their own workplace. They will also discuss and generate ideas how to follow-up on the preventive care process on an organisational level. This part of the intervention is intended to inspire end users to prevent pressure ulcers, malnutrition, poor oral health and falls among older persons in nursing homes.

### The format of the intervention
#### Part one of the intervention: the foundation and stairs 1–5
From end users' perspective, it was important with a clear format. They expressed a need of a structured, readable and colourful working 'manual'. Hence, the format of the intervention is designed as colourful stair with the intention to visualise the entire preventive working process. To enhance the structure, end users are provided with written instructions in respective stair. Furthermore, stakeholder emphasised the need of a 'self-check box' for end users when completing the foundation and stairs in the intervention. Stakeholder believed that this could increase participation and involvement among end users. Since both end users and stakeholders stressed that there are many end users who do not have the Swedish language as their native language, the language is adjusted to suit the local context. Furthermore, end users expressed that the format of the intervention should consider different ways of learning. This was also highlighted as important by stakeholders. Hence, the format consist of both reading texts and watching videos. Moreover, end users and stakeholders emphasised that a digital intervention could be a sustainable solution.

#### Part two of the intervention: stair 6
End users and stakeholder were in agreement that it is necessary to meet and discuss. Therefore, in part 2 of the intervention, end users meet in their respective nursing home. Also, the format of this part of the intervention was designed as an inspiration to raise awareness of the preventive work among end users.

### The delivery of the intervention
#### Part one of the intervention: the foundation and stairs 1–5
The intervention will be delivered via email to managers in nursing homes. Subsequently, respective manager will forward the intervention via workplace email addresses to nurse aides and registered nurses. The end users highlighted that some learn better individually, while others learn better in group. Therefore, they are permitted to choose if they want to read texts and watch videos individually and/or in group. *The foundation and stairs 1–5* is anticipated to take approximately 10 min, 60 min, 20 min, 10 min, 30 min and 60 min, respectively, for end users to complete. End users can choose to complete this part of intervention at once or divide it during working hours.

#### Part two of the intervention: stair 6
Part two of the intervention will be delivered by a facilitator (MN) who will moderate sessions lasting approximately 30 min each, Monday–Friday, in the nursing

homes in question. If end users participate in all the sessions during this week, the planned amount of time is two and half hours for each end user.

## DISCUSSION

The current codesigned complex intervention, the STAIR OF KNOWLEDGE, was developed together with end users in workshops in an active and creative way. Stakeholders were also engaged in an iterative and dynamic way throughout the development of the intervention, as an important part of undertaking implementation strategies already in the development phase.[39] As recommended by the MRC framework,[24] we meticulously considered the relationship between the intervention and its context when developing the intervention. Furthermore, we followed the strategies for knowledge translation included in the KTA framework.[18] Hence, the strengths exhibited by the development of this complex intervention lie in the fact that it was developed both together with and for end users and engaged stakeholders who are considered to play an important role in the development and implementation process. The current intervention is intended to work in a real-world setting and aims to bridge the evidence–practice gap regarding the process of preventing the risks of pressure ulcers, malnutrition, poor oral health and falls; ultimately, this intervention may reduce these risks among older persons in nursing homes.

When developing new intervention, the value of the used design process cannot be understated.[40] In fact, engagement of end users in a creative environment has been linked to more robust research and development efforts, which in turn may drive more successful interventions outcome.[40] Hence, the benefits of codesign are potentially substantial.[41] For instance, engaging end users and stakeholders as design partners to the research group could ensure that the intervention exhibits a better fit to their needs.[32] Engaging end users and stakeholders early enables their experiences and requirements to be taken into account at the start rather than a situation in which the researchers presume to know what is required.[39] In the current development process, although end users' and stakeholders' engagement ranged in intensity from relatively passive to highly active, their engagement pervaded the entire development process and important decisions regarding the intervention design were made by considering their input. Furthermore, because we engaged end users and stakeholders, the current intervention was based on their own experiences regarding the evidence and knowledge that are necessary throughout the entire process of preventing the risks of pressure ulcers, malnutrition, poor oral health and falls. Engaging end users and stakeholders during the developing process[42] was also important in light of the local context since this enabled us to identify facilitators and barriers in the environment in which the intervention will eventually be implemented.[26]

A recent scoping review investigating education interventions for health professionals on fall prevention in healthcare settings[43] highlighted that health professional education to prevent fall is important. Nevertheless, the scoping review concluded that there are no uniform education design principles utilised to date.[43] Another review found that it was uncertain whether education delivered in different formats such as didactic or video-based format makes a difference to health professionals' knowledge of pressure ulcers prevention. However, education format in the current developed intervention was designed to fit end users' needs and suit the local context, which may have benefits for the outcome.

Considering and understanding the local context is also crucial when addressing an evidence–practice gap.[24] In this case, knowledge concerning the process of preventing the risks of pressure ulcers, malnutrition, poor oral health and falls is already contained in the quality register Senior Alert, but this evidence has not been fully translated into practice. Thus, we focused on translating the existing knowledge contained in Senior Alert into practice. However, if this knowledge is to be implemented effectively,[44] it is crucial to employ a conceptual framework.[20] Therefore, we chose the KTA framework because it provided us with knowledge translation strategies to reduce the evidence–practice gap,[18] and it was suitable since the quality register Senior Alert is already in use. Furthermore, adapting knowledge to the local context and assessing barriers to knowledge use may enable the research to have a greater impact,[45] which could in turn reduce the evidence–practice gap.

Successful intervention development is characterised as rigorous and scientific and leads to an intervention that can be implemented in a real-world setting.[33] The robust research process used to develop the STAIR OF KNOWLEDGE intervention incorporates existing evidence, the views of end users and stakeholders,[41] the local context and knowledge translation strategies. Consequently, the use of knowledge translation strategies and the engagement of end users who are embedded in the local context in the development of a tailored complex intervention both for and with them could contribute to increased knowledge and awareness of the entire process of preventive care. This may, in turn, reduce the evidence–practice gap among end users and, importantly, reduce the risk of pressure ulcers, malnutrition, poor oral health and falls among older persons in nursing homes. Furthermore, the engagement of stakeholders already in the development process is likely to facilitate the implementation of the current intervention.

### Limitations

Although the development of this complex intervention has been completed, it is important to acknowledge the limitations of the development process. First, only four clusters were included in the development process. Nevertheless, since this part of the trial focused on the development of an intervention rather than its evaluation

and because the clusters were recruited pragmatically, the clusters included in the trial could be considered sufficient. Second, although all end users in the intervention arm (n=118) were invited to participate in workshops, only 16 participated. However, different professionals participated in the workshops, and the discussions were energetic, active and creative. Third, although this design is creative and can generate new ideas, it is time-consuming and rescore-consuming for all parties involved. It requires end users and stakeholders to set aside time and expend extra effort in their daily work. For researchers, this process requires careful planning to enable them to coordinate, meet with many different persons repeatedly and be responsive to all parties involved. However, although this design required the expenditure of time and resources, the engagement of end users, stakeholders and researchers is meaningful and necessary to develop successful interventions; ultimately, this design might have an impact on to prevent the risks of pressure ulcers, malnutrition, poor oral health and falls among older persons in nursing homes. Furthermore, the current intervention might offer value when used by others and could likely be adjusted to and tested in similar contexts.

## CONCLUSION

The current codesign complex intervention, the STAIR OF KNOWLEDGE, which aims to prevent the risks of pressure ulcers, malnutrition, poor oral health and falls among older persons in nursing homes, is robustly developed and thoroughly described. A careful description of the entire development process and the intervention itself can enhance the replicability of the current intervention. This article highlights the extensive process that is necessary for the development of tailored complex interventions. Finally, this codesigned complex intervention might result in more evidence-based practice concerning the entire process of preventing the risks of pressure ulcers, malnutrition, poor oral health and falls and, importantly, reduce these health risks among older persons in nursing homes. However, uncertainties regarding the intervention itself remain. Thus, the STAIR OF KNOWLEDGE must be tested and evaluated in an upcoming feasibility study before we continue to the stage of conducting a full trial evaluation.

## Ethical considerations

This trial was approved by the Swedish Ethical Review Authority (DNR 2019-06414). In addition, written approval was requested and granted by the head of the department of elderly care homes in the municipality in which this trial was conducted. All end users working in eligible nursing homes were invited to participate in the workshops. Moreover, end users had the right to withdraw from participation at any stage without providing reasons and bearing any consequences. Participation in the workshops was based on written consent. The results of this trial may be considered to contribute to scientific value on good ethical grounds, and the benefits of participating in the trial outweigh the corresponding risks.

**Author affiliations**
[1]Department of Care Science, Faculty of Health and Society, Malmo University, Malmo, Sweden
[2]Department of Cardiothoracic and Vascular Surgery, Skane University Hospital, Skanes universitetssjukhus Malmo, Malmo, Sweden
[3]Health and Social Care, Strategic Development, Unit of Research and Development and Competence Centre, Malmö, Sweden

**Acknowledgements** We thank the nurse aides, registered nurses and managers who participated in the development of the intervention. We also thank the stakeholders who participated during this process. Their enthusiastic participation was helpful in developing the STAIR OF KNOWLEDGE intervention. The Derbring and Stölten foundation is acknowledged for its financial support.

**Contributors** MN: conceptualisation, methodology, investigation, writing—original draft, writing—review and editing, validation, formal analysis, visualisation, guaranter. MAx, CK and PH: conceptualisation, methodology, investigation, writing—review and editing, validation, supervision. MA: conceptualisation, methodology, investigation, writing—review and editing, validation, formal analysis, supervision. All the authors read and approved the final version of the manuscript.

**Funding** This research received finanical support with regard to conducting the workshops from the Derbrings and Stöltens research and development foundation. The funders had no role in the research manuscript's design, conduct, analysis, interpretation or drafting.

**Competing interests** None declared.

**Patient and public involvement** Patients and/or the public were involved in the design, or conduct, or reporting or dissemination plans of this research. Refer to the Methods section for further details.

**Patient consent for publication** Not required.

**Ethics approval** This study involves human participants. This study was approved by the Swedish Ethical Review Authority (DNR 2019-06414). Participants gave informed consent to participate in the study before taking part.

**Provenance and peer review** Not commissioned; externally peer reviewed.

**Data availability statement** No data are available. All data relevant to the study are included in the article or uploaded as supplementary information. The data that support the development of the STAIR OF KNOWLEDGE intervention are not publicly available to ensure confidentiality. All data relevant to the development are included in the article. All figures and tables included in this article are original.

**ORCID iD**
Merita Neziraj http://orcid.org/0000-0002-8169-853X

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
