## [Reviewer comments · BMJ Open]

ARTICLE DETAILS

TITLE (PROVISIONAL)	The STAIR OF KNOWLEDGE – a codesigned intervention to prevent pressure ulcers, malnutrition, poor oral health and falls among older persons in nursing homes in Sweden: development of a complex intervention
AUTHORS	Neziraj, Merita; Axelsson, Malin; Kumlien, Christine; Hellman, Peter; Andersson, Magdalena

VERSION 1 – REVIEW

REVIEWER	Balzer, Katrin Universität zu Lübeck, Institute for Social Medicine and Epidemiology
REVIEW RETURNED	26-Feb-2023

GENERAL COMMENTS	Thank you very much for the opportunity to review this manuscript on the development of a quality improvement intervention to advance the prevention and management of pressure ulcers, malnutrition and other typical age-related health risks in nursing home residents. The subject of this paper is clinically highly relevant as in many countries the prevalence of respective health problems in nursing home residents remains unchanged at high levels while epidemiological data indicate insufficient application of evidence-based recommendations for this population. The intervention newly developed by the authors aims to facilitate the translation of this knowledge into healthcare practice. The authors used up-to-date methods of participatory research and appropriate theoretical and methodological frameworks to co-create and co-design a knowledge-translation intervention that fits to the different groups of professionals involved in the delivery of nursing care, i.e. nursing aides, registered nurses and others. Overall, the manuscript is well reported and largely meets existing standards of reporting on intervention development for health service research (GUIDED reporting statement, Duncan et al. 2019, DOI: http://dx.doi.org/10.1136/bmjopen-2019-033516). However, at some points, the reproducibility of the methods and results is limited due to lacking information, and the linkage between the results of the single study parts and the eventually developed intervention remains vague. Major comments (1) Involvement of stakeholders, end-users and other perspectives *Although the authors provide a definition of the term “stakeholders”, it becomes not entirely clear how many stakeholders were involved at the different development steps and
---

which perspectives (i.e. professions/positions/roles and organizations/institutions) were represented by them. This should be clarified for each step where stakeholders participated.

*Also, the composition of the research and the reference groups should be described in more detail regarding the professions/positions/roles and organisation represented by the members of these groups. This description could be included below the headings "Study context and setting".

*Did involved stakeholders also include informal caregivers/relatives of the nursing home residents? Usually, these individuals would like to be integrated into the care for their family members, also in the nursing home, and may play a significant role in the application of evidence-based care recommendations.

*Altogether, I would welcome a figure or table that displays how many representatives of each group were planned to be/were actually involved in each phase of this intervention development study. This figure/table (for example to be inserted under the heading "Patient and public involvement") would be very useful to follow the co-design procedures and would also provide a more replicable indication of the amount of human resources invested in this development process. Since figure 4 only demonstrates the succession of iterations but without reference to the study phases and with lacking information about the composition of the research and references groups, it is not as informative as it should be.

(2) Study design and methods

*What is the relation between this three-phase development study and the PROSENIOR project. What are the aims and design of the PROSENIOR project? Is it the two-arm pragmatic cRCT conducted in phase 2 of this intervention development study? Or the name of the registry built up for quality measurement and research? If available, a reference to the protocol of this PROSENIOR project should be added (p 6, lines 37-39).

*Is there an a priori developed protocol of this intervention development study? If ever possible, a reference to this protocol should be included. It does not become entirely clear to which degree methods for this interventions development have been pre-planned or adapted/developed post-hoc in the course of this three-phase study. The authors should make transparent for each phase which methods had been preplanned and which ones had been added/amended post-hoc.

*Phase one, search for literature: Why did the authors search for studies on the risk of pressure ulcers etc.? This is a quite broad subject, and there is no straightforward linkage between this subject and the knowledge translation intervention to be developed. To make better clear which bodies of research evidence were considered in the development of the intervention, I strongly recommend that the authors add a supplementary file that provides detailed information on the aims, review questions, review methods and findings for all literature searches carried out for this intervention development study. For example, I would have expected that the authors systematically searched for up-to-date evidence on context factors facilitating/hindering implementation of evidence-based practice in long-term care institutions for the elderly.

*Phase 2, cRCT: Unfortunately, it appears very difficult to follow the methods of this phase. Was the cRCT specifically conducted for this phase of the development study? And what kind of intervention were the nursing homes randomised to? Generally, this cRCT has to be reported in more detail: objective, population, allocated interventions, outcomes and outcome measurement, time points, target sample size (nursing homes, persons), analysis methods. Is there a protocol of this trial which can be referred to here for further details? Also, it has to be described more plainly what the role of this cRCT was for the development of the target knowledge translation intervention and the workshops that were also conducted in phase 2.

*Phase 2, workshops: What has been the target sample size a priori: how many workshops, how many participants per relevant sub-group? How were the participants recruited?

*Phase 2, qualitative data analysis: The time frame of this analysis in relation to the time points of data collection should be reported. Were the data analysed parallel to the data collection? My understanding from table 1 (workshop plan) is, that the workshop participants already designed some core elements of the intervention, but later in the data analysis methods (p 11, line 38) it is reported that one author designed this during the data analysis. This reads confusing.

*Phase 3: Which methods were used for this phase? How many meetings were preplanned, how many were eventually conducted, involving how many persons from which group and how much time in total? Were the meetings structured and documented? And how did the research evidence retrieved in the earlier phases, e.g. literature searches and registry studies, feed into these iterations?

(3) Results

*This chapter includes a brief description of the final design, content and methods of the knowledge translation information but it remains unclear why the intervention was designed as such. For example, it seems that large parts of the intervention relies on self-directed learning by nursing aides, nurses and other staff members, which strikes me as one rationale for the development of the intervention was the recognition that it appears difficult to these staff members to keep track on available digital knowledge resources in routine care. How is this barrier addressed by the newly developed intervention?

*Also, the current description lacks some important information about the characteristics of the intervention, e.g. about the training/learning methods and didactic considerations for each stair and the amount of time planned for the staff members to spend at each stair. Also, re

*To overcome the missing link between study findings and intervention design/content and improve the reproducibility of intervention characteristics, I recommend that the authors complete the TIDieR checklist or the Cre-DEPTH checklist (van hecke et al. 2020, DOI: 10.1016/j.nedt.2019.104254) for this newly developed intervention and add this checklist as supplementary material to the results chapter. Each of these two checklists require that the authors justify the intervention content and methods chosen.

*Also, information about the staff and other resources required for implementation of this intervention and planned reimbursement sources should be added. For example, how will be the training activities of the author MN reimbursed when the intervention is due to routine implementation?

(4) Ethical considerations

Given the significant amount of time and thoughts spent by end-users, stakeholders and others on the intervention development, information about incentives/reimbursements provided to these target groups to acknowledge and appreciate their efforts and inputs should be added.

Minor comments

(5) General comment: At several instances, the authors write “to prevent the risk of pressure ulcers ...”. However, the risk of the health problems of interest cannot be prevented as it exists always but it has to be reduced or minimised.

(6) Title: The name of the country (Sweden) could be removed as the subject (need for effective knowledge translation methods) and the methods of the newly developed intervention may be relevant to many countries. Instead, the methodology of this research should be mentioned in the second part of the title, e.g. as follows: Development of a knowledge-to-action (STAIR OF KNOWLEDGE) intervention to prevent pressure ulcers, malnutrition, poor oral health and falls among older persons in nursing homes – a mixed methods study using co-design methods

(7) Abstract:

*Design: Currently, this para does not properly reflect the actual design of this three-phase study. The references to the MRC framework and the application under real word conditions appear not necessary for this abstract. Instead, the actual methods used should be briefly summarised.

*Participants: The number of representatives of each target/stakeholder group involved in this intervention development study should be mentioned.

(8) Methods

*Study context and setting: Please provide information about the number of nursing homes located in the target region (municipality) as well as the number of those nursing homes that participated in each of the three research phases.

*Phase one, conducting studies in the local context:

**How many focus groups with how many end-users were conducted? Please add this information.

**P 8, lines 25-27: What does the sentence “These data were not specifically targeted at the current part of the trial ...” mean? Again, it is not clear, which study was conducted for which purpose. This needs to become clearer throughout the whole manuscript.

*Figure 1: For each phase, the exact time period should be mentioned.

REVIEWER	Abraham, Jens Martin Luther University Halle-Wittenberg Institute of Health and Nursing Sciences, University Halle - Germany
REVIEW RETURNED	09-Mar-2023

GENERAL COMMENTS	Dear authors, Thank you for the opportunity to review the paper titled "Development of a codesigned complex intervention to prevent the risks of pressure ulcers, malnutrition, poor oral health and falls among older persons in nursing homes in Sweden - The STAIR OF KNOWLEDGE". In general, the intention to publish the development process of a complex intervention is very important and may support its implementation into practice. However, I have some comments and recommendations regarding this manuscript that should be considered. . Abstract The description of methods is partially insufficient. When were the workshops conducted? How were the data analyzed? In which way were the participants recruited?, etc. The results should also be described in more detail. How many workshops were held? How many people participated? etc. In contrast, the part regarding the aims of the complex intervention in the conclusion (P. 2, lines 35 & 36) is redundant with the objectives and should be avoided in favor of a more detailed description of the methods and results. Limitations In my opinion, the transferability of the results to other nursing homes as described in the limitations section (P. 15, lines 8 & 9) is a greater limitation than the time- and resourceconsuming process. Introduction P. 4, line 36: The comma after the reference 12 should probably not be superscript. Development of the intervention In Figure 1, phase 1 is a small typo (littérature). P. 9, line 10: Figure 2 should be referred to earlier (P. 8, line 57), as only the clusters and not the participants are shown here. P. 9, lines 22-25: Results are already described here that should be reported in the results section. Results In my opinion, the results section is quite brief. A description of the participants is missing (see previous comment), as well as the number and duration of the workshops and focus group interviews. The contents and topics could also be described in more detail and anchor examples could be included for better comprehensibility. Discussion Here, the results should also be comprehensively discussed in comparison to other comparable interventions aiming at prevention of the risks of pressure ulcers, malnutrition, poor oral health and falls (or some of these problems) in nursing homes or in other settings.
---

VERSION 1 – AUTHOR RESPONSE

Prof. Katrin Balzer, Universität zu Lübeck

The subject of this paper is clinically highly relevant as in many countries the prevalence of respective health problems in nursing home residents remains unchanged at high levels while epidemiological data indicate insufficient application of evidence-based recommendations for this population. The intervention newly developed by the authors aims to facilitate the translation of this knowledge into healthcare practice.

The authors used up-to-date methods of participatory research and appropriate theoretical and methodological frameworks to co-create and co-design a knowledge-translation intervention that fits to the different groups of professionals involved in the delivery of nursing care, i.e. nursing aides, registered nurses and others. Overall, the manuscript is well reported and largely meets existing standards of reporting on intervention development for health service research (GUIDED reporting statement, Duncan et al. 2019). However, at some points, the reproducibility of the methods and results is limited due to lacking information, and the linkage between the results of the single study parts and the eventually developed intervention remains vague.

Authors' reply: Dear prof. Balzer, thank you for your valuable comments. Below we have addressed your comments point-by-point. We think that your comments helped us improve our manuscript.

1. Involvement of stakeholders, end-users and other perspectives:

*Although the authors provide a definition of the term "stakeholders", it becomes not entirely clear how many stakeholders were involved at the different development steps and which perspectives (i.e. professions/positions/roles and organizations/institutions) were represented by them. This should be clarified for each step where stakeholders participated.

Authors' reply: Thank you for raising this question. We apologise for being unclear in this regard. In line with your suggestion, we clarified this in the manuscript. Please see: "Phase one" on page 8 that starts with: As a part of steps 2-3 in the KTA framework..., "Phase two" on page 10 in the first paragraph, "Results" on page 13 in the first paragraph and Figure 4.

*Also, the composition of the research and the reference groups should be described in more detail regarding the professions/positions/roles and organisation represented by the members of these groups. This description could be included below the headings "Study context and setting".

Authors' reply: Thank you for your suggestion. This information is now added in the paragraph "Study context and setting" on page 6.

*Did involved stakeholders also include informal caregivers/relatives of the nursing home residents? Usually, these individuals would like to be integrated into the care for their family members, also in the nursing home, and may play a significant role in the application of evidence-based care recommendations.

Authors' reply: Thank you for this important question. As you imply, informal caregivers are very important and play a significant role, however, in this study informal caregivers were not included since this part of the project focused on translating knowledge among healthcare staff and managers.

*Altogether, I would welcome a figure or table that displays how many representatives of each group were planned to be/were actually involved in each phase of this intervention development study. This figure/table (for example to be inserted under the heading "Patient and public involvement") would be very useful to follow the co-design procedures and would also provide a more replicable indication of the amount of human resources invested in this development process. Since figure 4 only demonstrates the succession of iterations but without reference to the study phases and with lacking

information about the composition of the research and references groups, it is not as informative as it should be.

Authors' reply: Thank you for this comment. Because this study is designed in a pragmatic paradigm, we did not anticipate participants beforehand. We planned for and conducted the development of the current intervention in a real-world setting, therefore, participation was based on the nursing home's ability to participate. We added this information in the paragraph "Patient and public involvement" on page 7. In line with your suggestion, we clarified the amount of human resources invested in this development process. Please see the paragraphs "Study context and setting" on page 6, "Phase one" on page 8, "Phase two" on page 10, "Results" on page 13 and Figure 4.

2. Study design and methods:

*What is the relation between this three-phase development study and the PROSENIOR project. What are the aims and design of the PROSENIOR project? Is it the two-arm pragmatic cRCT conducted in phase 2 of this intervention development study? Or the name of the registry built up for quality measurement and research? If available, a reference to the protocol of this PROSENIOR project should be added (p 6, lines 37-39).

Authors' reply: We apologise for being unclear in this regard. The overall aim and design of the PROSENIOR is clarified in the paragraph "Study design" on page 6. The purpose of the randomization is clarified in paragraph "Phase two" on page 9. There is a protocol available in clinical trial database (Clinical Trial NCT05308862).

*Is there an a priori developed protocol of this intervention development study? If ever possible, a reference to this protocol should be included. It does not become entirely clear to which degree methods for this intervention development have been pre-planned or adapted/developed post-hoc in the course of this three-phase study. The authors should make transparent for each phase which methods had been preplanned and which ones had been added/amended post-hoc.

Authors' reply: Thank you for your comment. The methods of the intervention development were carefully planned, considered and described in an available protocol in the clinical trial database (Clinical Trial NCT05308862).

*Phase one, search for literature: Why did the authors search for studies on the risk of pressure ulcers etc.? This is a quite broad subject, and there is no straightforward linkage between this subject and the knowledge translation intervention to be developed. To make better clear which bodies of research evidence were considered in the development of the intervention, I strongly recommend that the authors add a supplementary file that provides detailed information on the aims, review questions, review methods and findings for all literature searches carried out for this intervention development study. For example, I would have expected that the authors systematically searched for up-to-date evidence on context factors facilitating/hindering implementation of evidence-based practice in long-term care institutions for the elderly.

Authors' reply: Thank you for your comment. An update of the current status of the literature regarding pressure ulcers, malnutrition, poor oral health and falls and intervention studies in this area was made. However, since we did not conduct a systematic literature review this could be regarded as a limitation. It was difficult to conduct a systematic literature considering all four health risks: pressure ulcer, malnutrition, poor oral health and falls and intervention studies. But, in line with your comment, we have clarified this in the manuscript. Please see the paragraph "Phase one" on page 8.

*Phase 2, cRCT: Unfortunately, it appears very difficult to follow the methods of this phase. Was the cRCT specifically conducted for this phase of the development study? And what kind of intervention were the nursing homes randomised to? Generally, this cRCT has to be reported in more detail: objective, population, allocated interventions, outcomes and outcome measurement, time points, target sample size (nursing homes, persons), analysis methods. Is there a protocol of this trial which can be referred to here for further details? Also, it has to be described more plainly what the role of this cRCT was for the development of the target knowledge translation intervention and the workshops that were also conducted in phase 2.

Authors' reply: We apologise for being unclear. The purpose of randomization in the current study is clarified in the paragraph "Phase two" on page 9. The cRCT will be reported in detail in our upcoming feasibility study.

*Phase 2, workshops: What has been the target sample size a priori: how many workshops, how many participants per relevant sub-group? How were the participants recruited?

Authors' reply: We considered the nursing homes' and end users' possibility to participate in line with conducting pragmatic research. We recruited end users in nursing homes allocated to the intervention arm to participate in workshops. We invited all end users in nursing homes allocated to the intervention arm, hence 118 end users were invited to workshops both via a digital information video and in written form. Sixteen end users ultimately participated in the workshops. This is described in the paragraph "Recruiting and randomizing nursing homes" on page 9.

*Phase 3: Which methods were used for this phase? How many meetings were preplanned, how many were eventually conducted, involving how many persons from which group and how much time in total? Were the meetings structured and documented? And how did the research evidence retrieve in the earlier phases, e.g. literature searches and registry studies, feed into these iterations?

Authors' reply: Thank you for the important comment. All meetings were structured and documented by the first author. This information is now added in the paragraph "Phase three" on page 12. However, all meetings were not pre-planned as it was difficult to consider which key persons was going to play an important role in the development in advance. As recommended by the MRC framework, we meticulously considered the relationship between the intervention and its context when developing the intervention. Because we developed the intervention in codesign, we involved and engaged persons who played an important role in the development process. The iterative and dynamic process in this phase is also in line with the knowledge-to-action framework. In phase one, we searched for literature regarding the risks of pressure ulcers, malnutrition, poor oral health and falls and intervention studies in this area and we conducted a cross-sectional study to determine the prevalence of the risks of pressure ulcers, malnutrition, poor oral health and falls in nursing homes in southern Sweden and we conducted focus group interviews with end users. This evidence in the earlier phases is incorporated in the content of the workshops and in the iterative process in phase 3. As you suggested, we added the number of persons and the amount of time of the meetings in Figure 4.

3. Results

*This chapter includes a brief description of the final design, content and methods of the knowledge translation information but it remains unclear why the intervention was designed as such. For example, it seems that large parts of the intervention relies on self-directed learning by nursing aides, nurses and other staff members, which strikes me as one rationale for the development of the intervention was the recognition that it appears difficult to these staff members to keep track on available digital knowledge resources in routine care. How is this barrier addressed by the newly developed intervention?

Authors' reply: In line with your questions, we have rewritten the paragraph "Results" on page 12. Your comment helped us improve the description of the intervention, thank you.

*Also, the current description lacks some important information about the characteristics of the intervention, e.g. about the training/learning methods and didactic considerations for each stair and the amount of time planned for the staff members to spend at each stair. Also, re

Authors' reply: Thank you for this good suggestion. In line with your suggestion, this information is added in the paragraph "Results" on page 13.

*To overcome the missing link between study findings and intervention design/content and improve the reproducibility of intervention characteristics, I recommend that the authors complete the TIDieR checklist or the Cre-DEPTH checklist (van hecke et al. 2020, DOI: 10.1016/j.nedt.2019.104254) for this newly developed intervention and add this checklist as supplementary material to the results chapter. Each of these two checklists require that the authors justify the intervention content and methods chosen.

Authors' reply: In line with your suggestion, we have added the TIDieR checklist as supplementary material to the result on page 13.

*Also, information about the staff and other resources required for implementation of this intervention and planned reimbursement sources should be added. For example, how will be the training activities of the author MN reimbursed when the intervention is due to routine implementation?

Authors' reply: The intervention needs to be tested for its feasibility, suitability, and effectiveness before implantation into routine practice. Therefore, any reimbursements are not considered.

4. Ethical considerations

Given the significant amount of time and thoughts spent by end-users, stakeholders and others on the intervention development, information about incentives/reimbursements provided to these target groups to acknowledge and appreciate their efforts and inputs should be added.

Authors' reply: This is stated in paragraph "Acknowledgements" on page 20.

5. General comment:

General comment: At several instances, the authors write "to prevent the risk of pressure ulcers ...". However, the risk of the health problems of interest cannot be prevented as it exists always but it has to be reduced or minimised.

Authors' reply: Thank you for this comment. If health care staff work more in line with the existing evidence to prevent these health risks in the nursing homes, these health risks can be prevented in the first place. Health care staff risk assess older persons in nursing homes and among some of them, these health risks are not present. However, in case these health risks occurs among older persons, they can be reduced or minimized, as you suggest.

6. Title: The name of the country (Sweden) could be removed as the subject (need for effective knowledge translation methods) and the methods of the newly developed intervention may be relevant to many countries. Instead, the methodology of this research should be mentioned in the second part of the title, e.g. as follows: Development of a knowledge-to-action (STAIR OF KNOWLEDGE) intervention to prevent pressure ulcers, malnutrition, poor oral health and falls among older persons in nursing homes – a mixed methods study using co-design methods

Authors' reply: Thank you for this good suggestion. We have changed the title inspired by your suggestion.

7. Abstract:

*Design: Currently, this para does not properly reflect the actual design of this three-phase study. The references to the MRC framework and the application under real word conditions appear not necessary for this abstract. Instead, the actual methods used should be briefly summarized.

Authors' reply: In line with your suggestion, we added this information. Please see the abstract on page 2.

*Participants: The number of representatives of each target/stakeholder group involved in this intervention development study should be mentioned.

Authors' reply: This is added accordingly in the paragraph "Participants".

8. Methods

*Study context and setting: Please provide information about the number of nursing homes located in the target region (municipality) as well as the number of those nursing homes that participated in each of the three research phases.

Authors' reply: This is added accordingly. Please see the paragraph "Setting and study context" on page 6.

*Phase one, conducting studies in the local context:

**How many focus groups with how many end-users were conducted? Please add this information.

Authors' reply: This is added accordingly on page 8.

**P 8, lines 25-27: What does the sentence "These data were not specifically targeted at the current part of the trial ..." mean? Again, it is not clear, which study was conducted for which purpose. This needs to become clearer throughout the whole manuscript.

Authors' reply: The data were specifically targeted at the current study. This means that we did not analyse and report this data in our previous study but we included, analysed and reported in the current study. But since this was not clear, we have tried to further clarify this on page 8.

*Figure 1: For each phase, the exact time period should be mentioned.

Authors' reply: This is added accordingly. Please see Figure 1.

Reviewer: 2

Dr. Jens Abraham, Martin Luther University Halle-Wittenberg Institute of Health and Nursing Sciences

Comments to the Author:

Dear authors,

Thank you for the opportunity to review the paper titled "Development of a codesigned complex intervention to prevent the risks of pressure ulcers, malnutrition, poor oral health and falls among older persons in nursing homes in Sweden - The STAIR OF KNOWLEDGE". In general, the intention to publish the development process of a complex intervention is very important and may support its implementation into practice.

However, I have some comments and recommendations regarding this manuscript that should be considered.

Authors reply: Dear Dr. Abraham, thank you for your comments. Below we have addressed your comments point-by-point. We think that your comments helped us improve our manuscript.

Abstract

The description of methods is partially insufficient. When were the workshops conducted? How were the data analyzed? In which way were the participants recruited?, etc.

Authors reply: Thank you for this comment. The description of methos is changed as you suggested. Please see the abstract on page 2.

The results should also be described in more detail. How many workshops were held? How many people participated? etc. In contrast, the part regarding the aims of the complex intervention in the conclusion (P. 2, lines 35 & 36) is redundant with the objectives and should be avoided in favor of a more detailed description of the methods and results.

Authors reply: In line with your suggestions, the result and the conclusion are revised. Please see abstract on page 2.

Limitations

In my opinion, the transferability of the results to other nursing homes as described in the limitations section (P. 15, lines 8 & 9) is a greater limitation then the time- and resourceconsuming process.

Authors' reply: We considered your suggestion and added this as a limitation on page 3.

Introduction

P. 4, line 36: The comma after the reference 12 should probably not be superscript.

Authors' reply: Thank you for noticing the comma. It is now removed.

Development of the intervention

In Figure 1, phase 1 is a small typo (littrature).

Authors' reply: Thank you for noticing the typo. It is now corrected.

P. 9, line 10: Figure 2 should be referred to earlier (P. 8, line 57), as only the clusters and not the participants are shown here.

Authors' reply: We rewrote this part and referred earlier to Figure 2. Please the paragraph "Phase two" on page 9.

P. 9, lines 22-25: Results are already described here that should be reported in the results section. Authors' reply: This is changed in line with your suggestion. Please see the paragraph "Results" on page 13.

Results

In my opinion, the results section is quite brief. A description of the participants is missing (see previous comment), as well as the number and duration of the workshops and focus group interviews. The contents and topics could also be described in more detail and anchor examples could be included for better comprehensibility.

Authors' reply: Thank you for your valuable comment. In line with your comment, we rewrote the results. Please see the paragraph "Result" on page 13-16. We believe that your comment improved this part of our manuscript.

Discussion

Here, the results should also be comprehensively discussed in comparison to other comparable interventions aiming at prevention of the risks of pressure ulcers, malnutrition, poor oral health and falls (or some of these problems) in nursing homes or in other settings

Authors' reply: Thank you for this good suggestion. We have discussed the current intervention in comparison to other intervention aiming at preventing falls and pressure ulcers. This improved the discussion. Please see the paragraph "Discussion" on page 17.

VERSION 2 – REVIEW

REVIEWER	Balzer, Katrin Universität zu Lübeck, Institute for Social Medicine and Epidemiology
REVIEW RETURNED	07-May-2023

GENERAL COMMENTS	Thank you very much for the opportunity to review the revised version of the manuscript BMJ-Open-2023-072453.R1. The revision reads very well, and almost all of my concerns raised in the first review round have been properly addressed. There are only a few, mostly minor remarks left which should be taken into account before the manuscript will be considered for being accepted for publication: (1) Study methods (cRCT) The role of the cluster-randomized trial (cRCT) in this study is still not clear to me. The authors have to make clear whether the cRCT was used as a vehicle for phase 2, or whether phase 2 (workshops of intervention development) was a major aim of this cRCT. Probably, it is not true that this cRCT "was only conducted to invite end users ... to codesign n intervention ..." (p 9, lines 31-34). The authors' reply to my previous comments and other instances in the manuscript suggest, that this cRCT was primarily set up to assess the feasibility of the newly developed study. I recommend that the authors briefly describe the original or main aims of the cRCT, followed by a more detailed description how this trial was used for the intervention development. If the cRCT was not only used for the intervention co-design but also for the evaluation of feasibility, the CONSORT flowchart has to be revised so as to it visualizes the whole trial including all steps of intervention development and consecutive feasibility evaluation, including pre-planned measuring points and outcomes.
--

	(2) Structure of the paper The description of the methods of phase 1 closes with a para on the results of this phase. This mix-up of methods and results reads irritating. While most of the results have already been published elsewhere, I suggest that the authors add a brief summary of the main findings of each phase to the beginning of the results chapter. This would also provide little more context for the description of the sample of phase 2 (end users) and 3 (stakeholders) which currently stands quite alone. Furthermore, such a brief summary would strengthen the plausibility and reproducibility of the STAIR OF KNOWLEDGE intervention programme. (3) Language/formal issues *The references have to be checked and corrected. For example, in the text (e.g. p 8) the reference #2 is mentioned for a focus group study by Neziraj et al., but in the reference list this study (reference #2) is listed with the authors Merita et al. (2021). *P 5, line 48: Here a definition of nursing home residents is provided, not of nursing homes (since nursing homes are not (older) people. *P 8, lines 19/20: Here, the syntax of the accessory clause (“updated us ...”) has to be corrected. *P 8, line 53: Since the study referenced #2 has already been introduced in the foregoing sentence, it is not necessary to repeat the authors of this study here again. It would be sufficient if just the correct reference number is mentioned at the end of this sentence. *Figure 3: To my perception, figure 3 does not add relevant information. Furthermore, as the text displayed in these photos is written in Swedish language, these illustrations may not be comprehensible at one glance for many users of this research. I suggest omitting of figure. Despite my remaining comments, I congratulate the authors for this rigorously developed intervention and do look forward to upcoming findings from the evaluation of this intervention.
--	---

REVIEWER	Abraham, Jens Martin Luther University Halle-Wittenberg Institute of Health and Nursing Sciences, University Halle - Germany
REVIEW RETURNED	14-Apr-2023

GENERAL COMMENTS	In general, the manuscript has been revised satisfactorily in my opinion. However, the new paragraph regarding the study design (page 8, lines 5 - 19) is difficult to follow and contains some repetitions. This should be checked again and revised if necessary. I wish you much success for your further submission.
---

VERSION 2 – AUTHOR RESPONSE

Reviewer: 1

Prof. Katrin Balzer, Universität zu Lübeck

Comments to the Author:

Thank you very much for the opportunity to review the revised version of the manuscript BMJ-Open-2023-072453.R1. The revision reads very well, and almost all of my concerns raised in the first review round have been properly addressed. There are only a few, mostly minor remarks left which should be taken into account before the manuscript will be considered for being accepted for publication.

Authors reply: Dear prof. Balzer, thank you for your comments. Below we have addressed your comments point-by-point.

(1) Study methods (cRCT)

The role of the cluster-randomized trial (cRCT) in this study is still not clear to me. The authors have to make clear whether the cRCT was used as a vehicle for phase 2, or whether phase 2 (workshops of intervention development) was a major aim of this cRCT. Probably, it is not true that this cRCT “was only conducted to invite end users ... to codesign n intervention ...” (p 9, lines 31-34). The authors’ reply to my previous comments and other instances in the manuscript suggest, that this cRCT was primarily set up to assess the feasibility of the newly developed study. I recommend that the authors briefly describe the original or main aims of the cRCT, followed by a more detailed description how this trial was used for the intervention development. If the cRCT was not only used for the intervention co-design but also for the evaluation of feasibility, the CONSORT flowchart has to be revised so as to it visualizes the whole trial including all steps of intervention development and consecutive feasibility evaluation, including pre-planned measuring points and outcomes.

Authors reply: Thank you for raising this question. It is true that the randomization was not only conducted to invite end users to codesign. The role of the randomization was primarily to assess the procedure and will be reported in our upcoming feasibility study. Furthermore, we did not specify the feasibility objectives in the current study since the focus of the current study was to develop an intervention. The role of the randomization in the current study served the purpose to invite end users in the intervention arm to workshops. In line with your suggestion, we clarified the purpose of the randomization for the whole trial and for the current study in the paragraph Study design on page 6-7, in the paragraph Recruiting and randomizing nursing homes on page 9 and the CONSORT flowchart, please see figure 2. We hope that the role of the randomization is clear now.

(2) Structure of the paper

The description of the methods of phase 1 closes with a para on the results of this phase. This mix-up of methods and results reads irritating. While most of the results have already been published elsewhere, I suggest that the authors add a brief summary of the main findings of each phase to the beginning of the results chapter. This would also provide little more context for the description of the sample of phase 2 (end users) and 3 (stakeholders) which currently stands quite alone. Furthermore, such a brief summary would strengthen the plausibility and reproducibility of the STAIR OF KNOWLEDGE intervention programme.

Authors reply: As you recommended a brief summary of phase one-three is added in the beginning of the results chapter. Please see page 13. Thank you for this comment, it improved our manuscript.

(3) Language/formal issues

*The references have to be checked and corrected. For example, in the text (e.g. p 8) the reference #2 is mentioned for a focus group study by Neziraj et al., but in the reference list this study (reference #2) is listed with the authors Merita et al. (2021).

Authors reply: Thank you for noticing. This is now changed.

*P 5, line 48: Here a definition of nursing home residents is provided, not of nursing homes (since nursing homes are not (older) people).

Authors reply: This is changed as you suggested. Please see page 5.

*P 8, lines 19/20: Here, the syntax of the accessory clause (“updated us ...”) has to be corrected.

Authors reply: This is corrected as you suggested. Please see page 8.

*P 8, line 53: Since the study referenced #2 has already been introduced in the foregoing sentence, it is not necessary to repeat the authors of this study here again. It would be sufficient if just the correct reference number is mentioned at the end of this sentence.

Authors reply: This is changed as you suggested. Please see page 8.

*Figure 3: To my perception, figure 3 does not add relevant information. Furthermore, as the text displayed in these photos is written in Swedish language, these illustrations may not be comprehensible at one glance for many users of this research. I suggest omitting of figure.

Authors reply: Figure 3 is omitted as you suggested.

Despite my remaining comments, I congratulate the authors for this rigorously developed intervention and do look forward to upcoming findings from the evaluation of this intervention.

Authors reply: Thank you for putting a lot of effort into this manuscript. Your comments were valuable and helped us improve our manuscript.

Reviewer: 2

Dr. Jens Abraham, Martin Luther University Halle-Wittenberg Institute of Health and Nursing Sciences
Comments to the Author:

In general, the manuscript has been revised satisfactorily in my opinion. However, the new paragraph regarding the study design (page 8, lines 5 - 19) is difficult to follow and contains some repetitions. This should be checked again and revised if necessary.

I wish you much success for your further submission.

Authors reply: Dear Dr. Abraham, thank you for taking time to review our manuscript.

In line with your suggestion, the study design paragraph is revised. Please see page 6-7.

VERSION 3 – REVIEW

REVIEWER	Balzer, Katrin Universität zu Lübeck, Institute for Social Medicine and Epidemiology
REVIEW RETURNED	14-Jun-2023

GENERAL COMMENTS	Review on manuscript bmjopen-2023-072453.R2 Thank you very much for the opportunity to review again this revised version. The reporting quality has once more benefited from the revision. Almost all of my previous comments have been addressed in plausible manner. There are only very few points left that should be specified more precisely in the manuscript. They all relate to the use of randomised allocation in this study: *P 7, lines 9-12, an p 10, lines 23-30: These two para are quite redundant as each describes that the current development study (phase 2) was embedded in a cluster-randomised controlled trial (cRCT) evaluating the feasibility “of the procedure2. However, the information given is not only redundant but also still vague or prone to misunderstanding in some points. Following aspects should be clarified:  - Apparently, the development of the STAIR OF KNOWLEDGE programme and feasibility evaluation were subject of the same two-arm cRCT. Thus, randomization was only conducted once (and not several times as suggested by the current wording in the manuscript) with the double aim to first develop the intervention and then to evaluate the feasibility in the nursing homes allocated to the intervention group. The current paper only reports on the intervention development part of this trial, the feasibility evaluation will be reported separately elsewhere. I suggest that the authors introduce the trial in exactly this way on p 7, lines 9-12 (subsection "study design"). To facilitate the understanding of the readers, a reference to the trial flowchart should already be included here when the trial is mentioned the first time. - Also, the unit (nursing homes) and methods of randomised allocation should already (and only) be specified when the cRCT is introduced at the beginning of the methods chapter (subsection "study design", p 7, lines 9-12). Here, the authors should briefly describe how and by whom the randomization sequence was generated and implemented. Furthermore, the authors should revise the sentence “... the cluster randomization of nursing homes was unblinded”. Does this sentence refer to the concealment of allocation? How and by whom were nursing homes informed about the allocation output? And what does the unblinding of the nursing homes mean? Who was unblinded against what? This has to be reported more precisely. - Furthermore, the authors should also describe more precisely which procedures were tested with regard to feasibility in part two of the cRCT (intervention arm). Is it the procedures of the STAIR OF KNOWLEDGE programme or the general trial procedures (recruitment, data collection etc.)?
--

	- At the beginning of the methods subsection on phase two of the intervention development (p 10, lines 23-30), it would be sufficient if the authors briefly repeat here that this phase was part one of the cRCT. If all basic information on this trial is reported in the subsection of Study Design, no additional information would be required here. I wish the authors much success for this remaining revision and their further research on this subject.
--	--

VERSION 3 – AUTHOR RESPONSE

Prof. Katrin Balzer, Universität zu Lübeck

Comments to the Author:

Review on manuscript bmjopen-2023-072453.R2 Thank you very much for the opportunity to review again this revised version. The reporting quality has once more benefited from the revision. Almost all of my previous comments have been addressed in plausible manner. There are only very few points left that should be specified more precisely in the manuscript. They all relate to the use of randomised allocation in this study:

*P 7, lines 9-12, an p 10, lines 23-30: These two para are quite redundant as each describes that the current development study (phase 2) was embedded in a cluster-randomised controlled trial (cRCT) evaluating the feasibility “of the procedure2. However, the information given is not only redundant but also still vague or prone to misunderstanding in some points. Following aspects should be clarified:

- Apparently, the development of the STAIR OF KNOWLEDGE programme and feasibility evaluation were subject of the same two-arm cRCT. Thus, randomization was only conducted once (and not several times as suggested by the current wording in the manuscript) with the double aim to first develop the intervention and then to evaluate the feasibility in the nursing homes allocated to the intervention group. The current paper only reports on the intervention development part of this trial, the feasibility evaluation will be reported separately elsewhere. I suggest that the authors introduce the trial in exactly this way on p 7, lines 9-12 (subsection "study design"). To facilitate the understanding of the readers, a reference to the trial flowchart should already be included here when the trial is mentioned the first time.

Authors reply: Thank you for this good suggestion. Please see the revision in line with your suggestion on page 7 on lines 2-14.

- Also, the unit (nursing homes) and methods of randomised allocation should already (and only) be specified when the cRCT is introduced at the beginning of the methods chapter (subsection "study design", p 7, lines 9-12). Here, the authors should briefly describe how and by whom the randomization sequence was generated and implemented. Furthermore, the authors should revise the sentence "... the cluster randomization of nursing homes was unblinded". Does this sentence refer to the concealment of allocation? How and by whom were nursing homes informed about the allocation output? And what does the unblinding of the nursing homes mean? Who was unblinded against what? This has to be reported more precisely.

Authors reply: This is revised in line with your suggestion on page 7 on lines 9-13.

- Furthermore, the authors should also describe more precisely which procedures were tested with regard to feasibility in part two of the cRCT (intervention arm). Is it the procedures of the STAIR OF KNOWLEDGE programme or the general trial procedures (recruitment, data collection etc.)?

Authors reply: This is added. Please see page 7 on lines 5-7.

- At the beginning of the methods subsection on phase two of the intervention development (p 10, lines 23-30), it would be sufficient if the authors briefly repeat here that this phase was part one of the cRCT. If all basic information on this trial is reported in the subsection of Study Design, no additional information would be required here.

Authors reply: Thank you for this good suggestion. Please see the revision in line with your suggestion on page 9 on lines 18-20.

I wish the authors much success for this remaining revision and their further research on this subject

Authors reply: Dear Prof. Katrin Balzer, thank you for taking your time to review our manuscript once more. Your suggestions improved our manuscript.

VERSION 4 – REVIEW

REVIEWER	Balzer, Katrin Universität zu Lübeck, Institute for Social Medicine and Epidemiology
REVIEW RETURNED	30-Jun-2023

GENERAL COMMENTS	All of the previous comments have been adequately addressed. There are only two minor editorial comments which may be considered by the authors when the manuscript will be prepared for publication: *P 8, line 3: While the authors implemented the wording suggested in ne of my previous comments, I wonder whether the wording should rather be changes into "The 3 randomization was conducted with the twofold aim ...". This wording sounds to me more appropriate than "double aim". P 10, line 12: It may be useful to readers if once again a reference to figure 1 (CONSORT flowchart) is included at the end of the sentence "During this phase, we recruited and randomized nursing homes." These comments are just for the authors' reconsiderations. To my perception, the manuscript now justifies being accepted for publication. I wish the authors successful dissemination of their work and look further to upcoming publications on the feasibility trial and future research building upon the developed intervention for knowledge translation in the long-term care setting.
--